# Impact of COVID-19Quarantine on Low Back Pain Intensity, Prevalence, and Associated Risk Factors among Adult Citizens Residing in Riyadh (Saudi Arabia): A Cross-Sectional Study

**DOI:** 10.3390/ijerph17197302

**Published:** 2020-10-06

**Authors:** Peter Šagát, Peter Bartík, Pablo Prieto González, Dragoș Ioan Tohănean, Damir Knjaz

**Affiliations:** 1Health and Physical Education Department, Prince Sultan University, Riyadh 12435, Saudi Arabia; sagat@psu.edu.sa (P.Š.); pbartik@psu.edu.sa (P.B.); pprieto@psu.edu.sa (P.P.G.); 2Faculty of Health, Catholic University in Ružomberok, 03401 Ružomberok, Slovakia; 3Faculty of Physical Education and Mountain Sports, Transilvania University of Brașov, 500036 Brașov, Romania; dragos.tohanean@unitbv.ro; 4Faculty of Kinesiology, University of Zagreb, 10000 Zagreb, Croatia

**Keywords:** low back pain, quarantine, prevalence, risk factors

## Abstract

This study aimed to estimate the effect of the coronavirus disease 2019 (COVID-19) quarantine on low back pain (LBP) intensity, prevalence, and associated risk factors among adults in Riyadh (Saudi Arabia). A total of 463 adults (259 males and 204 females) aged between 18 and 64 years and residing in Riyadh (Saudi Arabia) participated in this cross-sectional study. A self-administered structured questionnaire composed of 20 questions regarding demographic characteristics, work- and academic-related aspects, physical activity (PA), daily habits and tasks, and pain-related aspects was used. The LBP point prevalence before the quarantine was 38.8%, and 43.8% after the quarantine. The LBP intensity significantly increased during the quarantine. The low back was also the most common musculoskeletal pain area. Furthermore, during the quarantine, a significantly higher LBP intensity was reported by those individuals who (a) were aged between 35 and 49 years old, (b) had a body mass index equal to or exceeding 30, (c) underwent higher levels of stress, (d) did not comply with the ergonomic recommendations, (e) were sitting for long periods, (f) did not practice enough physical activity (PA), and (g) underwent teleworking or distance learning. No significant differences were found between genders. The COVID-19 quarantine resulted in a significant increase in LBP intensity, point prevalence, and most associated risk factors.

## 1. Introduction

At present, the problems associated with low back pain (LBP) represent a major concern for public health authorities, as well as for the general population in developed countries [1]. Worldwide, it has been estimated that the LBP prevalence ranges from 1.4 to 20.0% [2]. LBP is, in fact, the most common cause of work-related musculoskeletal disorders in certain regions [3], and it also causes significant problems in both the personal and professional lives of individuals. This includes sleeping disorders, disability, invalidity, work absenteeism, lack of productivity, and difficulties in carrying out the profession chosen by each worker [4]. The economic impact of LBP also represents a big concern worldwide. In Western countries, it has been estimated that the costs of back pain range between 1 and 2% of the gross national product [5]. In the United States, experts have calculated that this condition’s total cost exceeds $100 billion per year [6].

The onset of LBP is often associated with the adoption of poor postures at work; heavy lifting; performing repetitive movements; trunk flexion, rotation, and hyperextension; pushing; pulling; carrying; whole-body vibrations [7]. In addition, certain factors can aggravate the LBP intensity, including age, gender, hypertension, smoking, ergonomics, lack of job satisfaction, being overweight or obese, lack of physical activity (PA), and depression [8,9]. Knowing these factors is essential because it is possible to design a prevention strategy once they are identified.

As for Saudi Arabia, the prevalence of LBP has been analyzed in recent studies. Most of them were done with specific groups, and it was found that the prevalence among nurses was 80% [10], 70% in dentists [3], 73.9% in health personnel [11], 68% among female secondary school teachers [12], and 57.3% among male high school teachers [13]. In contrast, only a few epidemiological studies have been conducted that aimed to analyze the prevalence of the general population in Saudi Arabia. In this sense, Awaji [14] found out in a recent review study that the LBP prevalence in this country ranges between 53.2% and 79.17%. However, the aforementioned prevalence levels may vary when the individual habits and lifestyle are modified. In this respect, the onset of the coronavirus disease 2019 (COVID-19) has forced many governments worldwide to make a series of decisions to prevent the pandemic´s rapid spread [15]. The precautionary measures implemented include social distancing, capacity limitations in public spaces and private homes, isolation, quarantine, and curfew enforcement [16]. Therefore, it is conceivable that all these events are likely to have affected people’s lives physically, emotionally, and psychologically. In fact, Mattioli et al. [17] state that quarantine measures have a negative impact on human beings in many aspects, which include (a) increased anxiety, anger, and stress; (b) decreased outdoor exercise and the overall amount of PA; (c) both stress and depression, which can lead individuals to adopt unhealthy dietary habits. Since many of these aspects are factors that worsen LBP, as explained before, it is conceivable that during the COVID-19 quarantine, the prevalence of this condition has increased. In this context, the present study´s purpose was to estimate the effect of the mentioned quarantine on LBP intensity, prevalence, and risk factors among adult citizens residing in Riyadh (Saudi Arabia). We hypothesized that (a) the prevalence of LBP, as well as its intensity among those citizens who already had this condition, has increased; (b) the factors aggravating LBP have undergone significant variations.

## 2. Materials and Methods

An analytical cross-sectional study was undertaken. 

### 2.1. Subjects

Adults (330 Saudi citizens and 133 foreigners; 259 males and 204 females; age: 35.63 ± 9.84 years) voluntarily participated in the current research. The inclusion criteria were (a) being aged between 18 and 64 years; (b) did not suffer from chronic psychological, physiological, or psychosomatic conditions; (c) were not hospitalized during the pandemic; (d) were a resident in Saudi Arabia; (e) stayed in Riyadh before and during the quarantine decreed by the Saudi authorities. All subjects received detailed information about the objectives, benefits, and risks associated with participation in this study. They also signed an informed consent form indicating their willingness to participate in the current research. The sample selection process was performed following the steps described in Figure 1 [18,19].

To assess the factors determining the presence of back pain, a self-administered structured questionnaire composed of 20 questions was used (Questionnaire S1, Appendix A). It was established that the following dimensions should be included: (a) demographic characteristics (age, gender, height, weight), (b) work- or academic-related aspects (the type of work or academic activity performed before and after the quarantine and type of activities performed while working or studying), (c) PA (type, frequency, duration), (d) daily habits and tasks (sitting, moving), (e) pain-related aspects (location and intensity before and after the quarantine), and (f) psychological aspects (stress level before and during quarantine). Equal importance was assigned to each item. To facilitate understanding the questionnaire, all items were written in simple, short, and plain language [20]. The questionnaire responses were structured on a scale of whole numbers from 1 to 5. By way of example, pain was rated from “no pain” to “extreme pain,” and stress was rated from “no stress” to “maximal stress.”

Before drafting the questionnaire, it was subjected to a validation process, as described in Figure 2 [18].

Subsequently, the reliability was also verified. For this purpose, a pre-trial was performed. Thirty subjects were asked to fill out the questionnaire. Then, the Cronbach alpha value was calculated by considering each item´s variances and the total variance [21]; the value obtained was α = 0.82, which reflected an adequate internal consistency. 

### 2.2. Questionnaire Application

Contact with potential study participants was established through the Riyadh municipality forum groups that were available on social media. Next, 1000 individuals were selected through a simple randomization process using SPSS software version 22.0 (SPSS, Inc., Chicago, IL, USA). Subsequently, the questionnaire was distributed among the selected citizens on 10 May 2020 at 8 a.m. and it was filled out anonymously. The collection of questionnaire responses finished on 17 May 2020 at 11.59 p.m. At that specific time, 811 responses had been received (81.1% response rate). Among these 811 respondents, 348 were ruled out because they did not meet the inclusion criteria. Therefore, the final sample was composed of 463 subjects. 

Additionally, it is very important to highlight the chronology of the curfew implementation in Riyadh´s city regarding the questionnaire dissemination timing. From the evening of 23 March 2020, a nationwide curfew planned for 21 days was implemented between 7 p.m. and 6 a.m. [22]. On 6 April 2020, a 24-hour curfew was announced. Movement was restricted to only essential travel between 6 a.m. and 3 p.m. The curfew´s sequential lifting started on 28 May 2020, until the total removal on 21 June 2020 [23]. 

### 2.3. Ethical Clearance

The study was conducted in accordance with the principles outlined in the Helsinki Declaration. It was also approved by the Institutional Review Board of the Bioethics Committee at Prince Sultan University in Riyadh, Saudi Arabia (approval no. 18/2020).

### 2.4. Statistical Analysis

All results are presented as mean (interquartile ranges). Kolmogorov–Smirnov and Levene’s tests were used to verify the normality and homoscedasticity, respectively. Since the data did not follow a normal distribution and the cohort sizes created to establish comparisons between specific conditions (i.e., gender, body mass index (BMI), age) was unequal, nonparametric tests were used. Therefore, comparisons of two sets of data were made using the Mann–Whitney *U* test, whereas the Kruskal–Wallis *H* test was conducted to make comparisons between more than two sets of data using Dunn–Bonferroni corrections. To make comparations between paired nominal data, McNemar’s test was conducted. Comparations of dichotomous dependent variables between three or more groups were made by using Cochran’s *Q* test with Bonferroni corrections. The Spearman test was used to calculate the correlation between variables, with the results being interpreted as follows: *r* = 0 null correlation, 0.01≤ *r* ≤ 0.09 very weak, 0.10 ≤ *r* ≤ 0.29 weak, 0.30 ≤ *r* ≤ 0.49 moderate, 0.50 ≤ *r* ≤ 0.69 strong, and *r* ≥ 0.70 very strong. To estimate the effect-size (ES), after applying the Mann–Whitney *U* test, the following formula was used: ES *= z*/√*N*. An ES of 0.2 was considered small, 0.5 moderate, and 0.8 large [24]. The percentage of change was calculated using the following formula: *%* change = ([final value − initial value]/initial value) × 100. The level of significance was set at *p* < 0.05. The statistical analysis was performed using SPSS software version 22.0 (SPSS, Inc., Chicago, IL, USA).

## 3. Results

The curfew decreed by the Saudi authorities implied the adoption of certain legal and institutional measures and mobility restrictions, which has impacted population habits and lifestyles. As shown in Table 1, the most prevalent musculoskeletal pain area was the low back, followed by the neck, shoulders, thoracic area, and legs during the quarantine. Furthermore, during confinement, the percentage of subjects who reported thorax and lower body pain significantly increased. Additionally, the individuals who indicated they did not suffer pain in any body area decreased but not significantly. The incidence of neck pain was clearly higher in women, whereas low back pain was fairly higher in men.

The confinement resulted in a significant increase in the percentage of the population carrying out teleworking and distance learning. Regarding the time spent sitting and moving, the number of respondents who were sitting all or most of the time during the quarantine significantly increased, whereas the percentage of interviewees who were moving always or most of the time significantly decreased. The cohorts of individuals who spent the same time sitting as moving experienced a slight decrease, which was not significant. 

As for PA, the percentage of subjects who did not practice PA and practiced only once a week significantly increased. Additionally, the number of individuals who practiced PA two, three, six, or seven times a week significantly decreased. Finally, during confinement, the percentage of subjects who reported more stress significantly increased. 

Furthermore, several comparisons were made between different sample cohorts and conditions (Table 2). In this way, it was observed that the LBP intensity reported by the subjects was significantly higher than before the quarantine (*p* < 0.001, ES = 0.18). However, no significant differences in LBP intensity were observed either before or during the quarantine between genders. Regarding the age, the 35-to-49-year-old cohort reported the higher LBP intensity, followed by the 50-to-64-year old cohort and the 18-to-34-year-old cohort before and during the quarantine. Significant differences were found between the 18-to-34-year-old cohort and the 35-to-49-year-old cohort before the quarantine (*p* < 0.001, ES = 0.29) and during the quarantine (*p* < 0.001, ES = 0.63), and between the 35-to-49-year-old and the 50-to-64-year-old cohort before the quarantine (*p* < 0.001, ES = 0.11) and during the quarantine (*p* < 0.001, ES 0.15). However, no significant LBP intensity differences were found between the 18-to-34-year-old and the 50-to-64-year-old cohort, either before or during the quarantine.

As for the BMI categories, the normal weight group reported a significantly lower LBP intensity than the overweight group before the quarantine (*p* < 0.001, ES = 0.54) and during the quarantine (*p* < 0.001, ES = 0.61), and than the obese group before the quarantine (*p* < 0.001, ES = 2.37) and during the quarantine (*p* < 0.001, ES = 2.38). Likewise, the overweight group reported lower pain than the obese group before the quarantine (*p* < 0.001, ES = 1.65) and during the quarantine (*p* < 0.001, ES = 2.38).

Individuals who suffered moderate or severe stress levels presented a significantly higher LBP intensity during the quarantine (*p* < 0.001, ES = 0.26) but not before. Furthermore, a significantly higher LBP intensity was observed among the subjects who did not comply with the ergonomic recommendations before (*p* < 0.001, ES = 2.39) and during the quarantine (*p* < 0.001, ES = 2.32).

Significant differences in the LBP intensity were observed between the individuals who underwent teleworking or online learning and the subjects who did not during the quarantine (*p* = 0.001, ES = 0.15) but not before the quarantine. Furthermore, those survey respondents who were moving always or most of the time reported a significantly lower LBP intensity, both before (*p =* 0.046, ES = 0.194) and during the quarantine (*p* < 0.001, ES = 0.188) than the individuals who were sitting all the time or most of the time.

Regarding the number of times per week the interviewees practiced PA, before the quarantine, the subjects who did not practice PA reported significantly higher LBP intensity than those who practiced PA four or five times a week (*p* < 0.001, ES = 0.75) and six or seven times a week (*p* = 0.007, ES = 0.89). Similarly, the individuals who practiced PA once a week reported significantly higher LBP intensity than those who practiced four or five times a week (*p* < 0.001, ES = 0.68) and six or seven times a week (*p* < 0.001, ES = 1.16). The cohort who practiced PA two or three times a week also reported a higher LBP intensity than the subjects who practiced four or five times a week (*p* < 0.001, ES = 0.49) and six or seven times a week (*p* < 0.001, ES = 0.99). No significant differences were found in LBP intensity between the cohort who practiced PA four or five times a week and the cohort who practiced PA six times a week or every day.

For during the quarantine, it was found that those individuals who did not practice PA presented a higher LBP intensity than those who practiced once a week (*p* < 0.001, ES = 0.52), two or three times a week (*p* = 0.019, *r* = 0.16), four or five times a week (*p* = 0.043, ES = 0.14), and six or seven times a week (*p* < 0.001, ES = 0.87). Similarly, the subjects who practiced PA once a week, presented a higher LBP intensity than the interviewees who practiced PA two or three times a week (*p* = 0.005, ES = 0.16), four or five times a week (*p* < 0.001, ES = 0.26), and six or seven times a week (*p* < 0.001, ES = 0.38).The LBP intensity reported by the cohort of respondents who practiced PA two or three times a week was also significantly higher than those who practiced PA six or seven times a week (*p* < 0.001, ES = 1.07). Finally, the cohort who practiced PA four or five times a week reported a significantly higher LBP intensity than the individuals who practiced PA six times a week or every day (*p* < 0.001, ES = 1.06).

The associations between the LBP risk factors were also estimated (Table 3). It was found that there was a significant positive correlation between the LBP intensity and time spent sitting during the quarantine, perceived stress before and during the quarantine, and BMI before and after the quarantine. A significant negative correlation was found between the weekly practice of PA during the quarantine and the LBP intensity. On the contrary, no significant correlation was observed between the LBP intensity and time spent sitting before the quarantine, weekly frequency of PA before the quarantine, compliance with ergonomic recommendations before and during the quarantine, and age before and during the quarantine.

## 4. Discussion

One of the present study´s main findings werethat the LBP´s point prevalence significantly increased after the lockdown, going from38.8% before the quarantine to 43.8% during the quarantine. Both figures are notably higher than the 23.8% LBP point prevalence observed by Alanzi et. al. [25] in a cross-sectional community-based study in the city of Arar (northern Saudi Arabia). The target population of both studies was composed of adults. The substantial differences between the city of Riyadh and Arar in terms of size and population (4,205,961 vs. 148,540 inhabitants) [26] could be the reason for this discrepancy. Another factor that might partially explain the lack of concordance between both studies is the increasing incidence of back pain over time in Saudi Arabia, as was observed in other countries [27]. In fact, Al-Arfaj et al. [28] reported a low back pain prevalence of 18.8% in 2003 in the region of Al-Qassim (Saudi Arabia), which would confirm the increasing tendency over time within the kingdom. However, our study´s LBP point prevalence was considerably lower than the 53.2% to 79.17% found by Awaji [14] in a review made using seven cross-sectional studies conducted in Saudi Arabia. In other recent studies also undertaken in Saudi Arabia among specific professional groups, the point prevalence of LBP found was 80% in nurses [10], 57.3% in male high school teachers [13], 55% among faculty members [29], 40.5% in medical students [30], 51.6% in taxi drivers, 31.4% in office workers [31], and 21.2% among health sciences students [32]. Hence, in most of these cases, the point prevalence was higher than in our study, which could be related to the burden of work, type of professional or academic activity carried out by each group, and poor posture at work [33]. Worldwide, the LBP´s point prevalence found in countries such as Canada, the United States, Sweden, Belgium, Finland, Israel, and the Netherlands ranges between 1.4 and 20.0% [3]. Therefore, the present study, and most of the studies conducted in Saudi Arabia, revealed a higher LBP point prevalence in Saudi Arabia than in foreign countries.

According to this study’s results, it was also possible to verify that the most common musculoskeletal pain area was the low back, followed by the neck. This result coincides with most of the existing studies conducted in Saudi Arabia that were related to musculoskeletal disorders [3,30,31,34]. However, this result was slightly different from the results found by Sirajudeen et al. [29] since they observed that the neck was the most common pain area, followed by the low back. Furthermore, it is also noteworthy that during the quarantine, the percentage of respondents who reported pain in all of the neck, shoulders, trunk, low back, and legs increased in all cases. In contrast, the percentage of subjects who did not present pain in any of the mentioned body areas decreased. 

The respondents´ average LBP intensity was significantly higher than before the quarantine, which reflected the negative effect of the restrictions undergone by individuals. As for gender, a higher prevalence of LBP was found in males both before and after the lockdown in our study. However, no significant differences were observed in back pain intensity between both genders. Although this result is consistent with the study conducted by Ferguson et al. [35] among manual material handling workers in the United States, recent studies have reported a higher LBP prevalence in women [3,9,34]. Therefore, it would be very useful to clarify the real impact of the gender factor on LBP in future research.

Moreover, the COVID-19 quarantine decreed by the Saudi authorities has caused significant changes in citizens´ lifestyles. While the number of times per week devoted to practicing PA decreased, the time spent sitting increased. Similarly, the percentage of individuals who reported more stress during the quarantine was much higher than those who suffered more stress before the lockdown. Consequently, it can be assumed that the alteration of these three factors increased the incidence of LBP. Thus, the subjects who were moving all the time or most of the time, the cohort who presented mild or no stress, and those who practiced PA with higher frequencies reported significantly lower LBP intensities (see Table 2). Similarly, the association between PA and LBP has been examined by Alzahrani et al. [36] through a meta-analysis, where they found a lower LBP prevalence among those individuals who regularly practiced PA. Taulaniemi et al. [37] found that exercise could reduce low back pain by improving lumbar movement control, abdominal strength, and physical functioning. 

The negative effect of prolonged sitting on LBP intensity verified in this study also coincides with the results obtained by Şimşek et al. [4]. Furthermore, Mörl&Bradl [38] suggest that extended periods of sitting implies the absence of lumbar muscle activation. This results in low conditioning of the low back muscles, which in turn overloads passive structures of this body area, such as intervertebral discs and ligaments. Similarly, our study´s results verified the negative effect of stress on LBP intensity, which is consistent with previous studies [10]. At this point, it is important to note that the perceived level of stress, both before and during quarantine, positively correlated with the LBP intensity, which reflected the relevant effect that this factor exerts in aggravating the pain. 

As shown in Table 1, the percentage of individuals who carried out teleworking or distance learning during the quarantine increased drastically. In this sense, significant differences in LBP intensity were reported by the subjects who conducted teleworking or distance learning during the quarantine but not before. A feasible explanation of this matter could be related to the burden of work or the study load that was undertaken. 

As demonstrated in previous research, BMI has proven to be an LBP risk factor [39]. In the present study, the normal weight cohort reported a significantly lower LBP intensity than the overweight and obese groups. Likewise, the overweight group reported lower pain than the obese group. This occurred because the excessive weight represented an additional overload for the spine structures. Excessive body weight can compress the spine and intervertebral discs, which may increase the risk of suffering nerve compression, disc and ligament degeneration, and impairment of the lumbosacral structures [40].

Interestingly, the 18-to-34-year-old age and the 50-to-64-year-old age cohorts reported a significantly lower LBP intensity than the 35-to-49-year-old age cohort. Although these results contradict some recent research [4,9], they are consistent with a study carried out by Shammari et al. [34], in which it was observed that the 30-to-39-year-old age group presented higher disabling musculoskeletal symptoms. Furthermore, this could be attributed to the higher workload and stress level that middle-aged adults undertook [41].

As for the correlations found between potential LBP risk factors and LBP intensity, stress and BMI had a significant positive correlation with LBP intensity before and during the quarantine, which reflected the evident effect of these two factors in aggravating LBP. Additionally, only during but not before the quarantine, there was a significant negative correlation between LBP intensity and PA, and a significant positive correlation between LBP intensity and time sitting. Therefore, these two factors exerted a clear influence during but not before the quarantine. In this way, it is possible to interpret that the LBP intensity increased mainly due to the variation in certain risk factors rather than their presence. In other words, it is conceivable that when people’s habits and routines undergo important alterations, their LBP intensity increases. However, this possibility has not been confirmed yet. Therefore, it might be clarified in future research.

Furthermore, no significant correlations were found between age and compliance with ergonomic recommendations with LBP intensity either before or after the quarantine. Regarding age, the lack of correlation can be explained because, in the current research, the adults aged between 35 and 49 years old reported a higher LBP intensity, as indicated previously. However, something slightly different seems to have happened with the adherence to ergonomic recommendations because, despite the significant differences observed between cohorts set by the degree of compliance with the mentioned recommendations, no significant correlation was found between this factor and the LBP intensity. Hence, in this specific case, it cannot be ruled out that those subjects who presented a higher LBP intensity had developed greater ergonomic awareness such that this circumstance may have increased the dispersion of variables observed when the correlation was calculated. The lack of correlation between the LBP intensity and certain LBP risk factors could also be attributed to the sample heterogeneity. Furthermore, it might reflect that, on the one hand, it is easy to identify LBP risk factors in small samples of specific groups (i.e., students, teachers, nurses), but on the other hand, it is difficult to identify those factors when evaluating larger samples or the general population. Hence, this aspect must be taken into account in epidemiological studies. 

Due to the complexity of the current research, to verify the association between LBP intensity and potential risk factors, a univariate analysis was used. Therefore, a possible joint variation of some of the mentioned risk factors cannot be excluded since a multivariate analytical approach was not adopted. 

In the present study, we assumed that increased sitting time and stress and decreased weekly practice of PA led to an increased LBP intensity since all the mentioned aspects are risk factors associated with LBP. However, it cannot be entirely ruled out that the sequence of events was the opposite. That is to say, the increase in LBP intensity might have been the cause but not the consequence of the decreased weekly practice of PA or increased stress. However, what has been verified was that the quarantine increased specific LBP risk factors and the prevalence of this musculoskeletal disorder. Therefore, it is necessary to take measurements to reverse this situation without delay. As such, a greater negative impact on adult citizens´ quality of life can be avoided. 

Finally, it is necessary to mention the limitations of the study. Due to the social distancing requirements, reduced mobility, and meeting restrictions, it was not possible to include certain measurements, such as inflammatory biomarkers and vitamin D levels, which could have provided relevant information regarding LBP risk factors, as observed in previous studies [42,43,44,45]. Furthermore, the LBP intensity was ascertained four weeks after the order of confinement. In this respect, it is necessary to recognize that some authors consider that pain recall is not entirely reliable [46], whereas other authors hold the opposite view [47]. Additionally, individuals suffering from chronic conditions and subjects that were hospitalized were not included in the current research. Thus, it was not possible to verify the confinement effect in this segment of the population.

## 5. Conclusions

The confinement decreed due to the COVID-19 pandemic led to a significant increase in LBP intensity among adults residing in Riyadh. Similarly, the LBP point prevalence increased from 38.8 to 43.8%. The low back was also the most common musculoskeletal pain area. Being aged between 35 and 49 years old, having a BMI equal to or exceeding 30, undergoing stress, non-adherence to ergonomic recommendations, prolonged sitting, the insufficient practice of PA, and undergoing teleworking or distance learning were associated with a higher LBP intensity.

## Figures and Tables

**Figure 1 ijerph-17-07302-f001:**
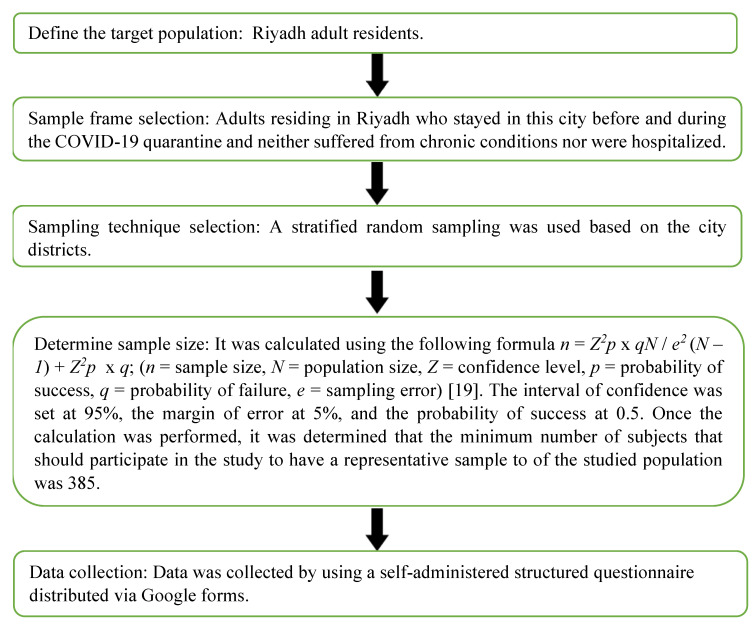
Sampling process steps.

**Figure 2 ijerph-17-07302-f002:**
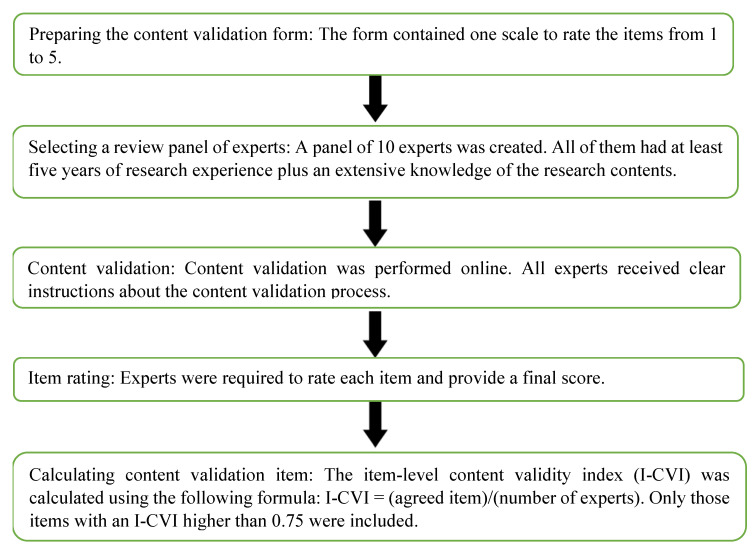
Questionnaire validation steps.

**Table 1 ijerph-17-07302-t001:** Changes experienced by citizens due to the quarantine.

Subjects	Body Area	Before the Quarantine * (%)	During the Quarantine # (%)	Percentage Change (%)	Significance Level (*p*)
Percentage of subjects who reported having pain in different body areas	Nowhere	35.2(F: 34.09, M: 36.31)	33(F: 33.33, M: 32.83)	−6.25	0.92
	Neck	25.8(F: 29.11, M: 21.89)	30.3(F: 32.56, M: 27.86)	17.44	0.17
	Shoulder(s)	18.5(F: 19.15, M: 17.91)	23.2(F: 23.37, M: 30.35)	25.41	0.21
	Thoracic area	9(F: 9.96, M: 7.96)	15.7(F: 15.32, M: 15.42)	74.44	0.02
	Low back	38.8(F: 35.25, M: 43.28)	43.8(F: 42.52, M: 44.7)	11.41	0.001
	Leg(s)	9.9(F: 10.34, M: 9.45)	13.9(F: 14.56, M: 13.43)	40.40	0.07
Percentage of subjects who did telework or distance learning		3.9(F: 47.14, M: 52.86)	48.3(F: 46.36, M: 51.25)	1138	<0.001
Percentage of subjects who were sitting always or most of the time		30.51(F: 32.43, M: 27.63)	50.9(F: 52.71, M: 50.02)	71.38	<0.001
Percentage of subjects who were sitting and moving equally		27.9(F: 23.94, M: 31.5)	24.2(F: 19.37, M: 28.57)	−13.26	0.24
Percentage who were moving always or most of the time		42.4(F:43.63, M: 40.5)	24.9(F: 27.92, M: 20.41)	−41.27	<0.001
Percentage of subjects who did not practice PA		7.3(F: 4.98, M: 9.41)	20(F: 22.23, M: 18.81)	173.97	0.001
Percentage of subjects who practiced PA once a week		10.3(F: 11.49, M: 8.49)	15.2(F: 14.50, M: 16.42)	47.57	0.02
Percentage of subjects who practiced PA two or three times a week		35.6(F: 39.84, M: 30.19)	25.1(F: 26.33, M: 23.38)	−29.49	0.001
Percentage of subjects who practiced PA four or five times a week		24.1(F: 24.52, M: 23.21)	25.8(F: 25.57, M: 25.94)	7.05	0.97
Percentage of subjects who practiced PA six or seven times a week		22.7(F: 19.17, M: 28.71)	13.9(F: 11.37, M: 15.65)	−38.76	<0.001
Percentage of subjects who reported a higher level of perceived stress		22.41(F: 52.99, M: 47.01)	50.43(F: 56.86, M: 43.13)	N/A	<0.001

* From 15 March to 22 March 2020; # From 10 May to 17 May 2020. F: female, M: male, PA: physical activity.

**Table 2 ijerph-17-07302-t002:** Pain intensity due to different factors, periods, and conditions.

Factor	Cohort	Pain Intensity before the Quarantine *	Pain Intensity during the Quarantine ▲
Time	Whole sample (*n* = 463)	1.95(2)	2.44(2) §
Gender	Male (*n* = 259)	1.96(2)	2.39(2)
	Female (*n* = 204)	1.95(2)	2.46(2)
Age group (years)	18–34 (*n* = 252)	1.90(2) +	2.35(2) +
	35–49 (*n* = 166)	2.04(2)	2.58(2)
	50–64 (*n* = 45)	1.93(2) +	2.44(2) +
Body mass index (BMI) category	Normal weight (*n* = 224)	1.93(2)	2.40(2)
	Overweight (*n* = 160)	1.97(2) †	2.44(2) †
	Obese (*n* = 71)	2.06(2) †, ††	2.64(2) †, ††
Perceived stress	Mild or no stress (*n* = 104)	1.94(2)	2.14(2)
	Moderate or severe (*n* = 234)	1.96(2)	2.73(2) ¥
Ergonomic recommendations compliance	Subjects who complied with the ergonomic recommendations (*n* = 63)	1.90(2)	2.27(2)
	Subjects who did not comply with the ergonomic recommendations (*n* = 223)	2.02(2) ■	2.63(2) ■
Carrying out teleworking or distance learning	No (*n* = 239)	1.94(2)	2.26(2)
	Yes (*n* = 224)	1.97(2)	2.64(2) ⅏
Time spent moving or sitting	Subjects who were sitting always or most of the time (*n* = 238)	2.11(2)	2.75(2)
	Subjects who were moving always or most of the time (*n* = 212)	1.92(2) #	2.23(2) #
Weekly practice of PA (times)	None (*n* = 91)	2.23(2)	2.98(2)
	1 (*n* = 71)	2.01(2)	2.75(2) &
	2–3 (*n* = 117)	1.93(2)	2.25(2) &, &&
	4–5 (*n* = 119)	1.87(2) &, &&, &&&	2.20(2) &, &&
	6–7 (*n* = 66)	1.81(2) &, &&, &&&	2.12(2) &, &&, &&&, &&&&

* From 15 March to 22 March 2020; pain was rated by the interviewees from 1 to 5, with 1 being no pain and 5 being extreme pain.▲ From 10 May to 17 May 2020; pain was rated by the interviewees from 1 to 5, with 1 being no pain and 5 being extreme pain. § Significant difference between both periods (before and during the quarantine), # significant difference from the age cohort who were moving always or most of the time, + significant difference from the 35–49-year-old age cohort, † significant difference from the normal weight cohort, †† significant difference from the overweight group, ¥ significant difference from the cohort that perceived mild or no stress, ■ significant difference from the cohort that complied with the ergonomic recommendations, ⅏ significant difference from the cohort that did not carry out teleworking or distance learning, # significant difference from the cohort that was moving always or most of the time, & significant difference from the cohort that did not practice PA, && significant difference from the cohort that practiced PA once a week, &&& significant difference from the cohort that practiced PA two or three times a week, &&&& significant difference from the cohort that practiced PA four or five times a week.

**Table 3 ijerph-17-07302-t003:** Correlations between back pain intensity and personal and environmental factors.

Factor	Back Pain Intensity before the Quarantine		Back Pain Intensity during the Quarantine	
Time spent sitting	*r* = 0.054	*p* = 0.216	*r =* 0.124	*p* = 0.008
Weekly frequency of PA	*r* = −0.023	*p* = 0.621	*r* = −0.198	*p* < 0.001
Perceived stress	*r* = 0.129	*p* = 0.014	*r* = 0.186	*p* < 0.001
Compliance with ergonomic recommendations	*r* = 0.030	*p* = 0.521	*r* = −0.059	*p* = 0.207
Age	*r* = 0.068	*p* = 0.147	*r =* −0.008	*p* = 0.869
BMI	*r* = 0.106	*p* = 0.029	*r =* 0.190	*p* = 0.009

*r*: Spearman correlation, *p*: significance level was set at <0.05.

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
