# Peer review of "Impact of COVID-19Quarantine on Low Back Pain Intensity, Prevalence, and Associated Risk Factors among Adult Citizens Residing in Riyadh (Saudi Arabia): A Cross-Sectional Study"

_ijerph, 2020, doi:10.3390/ijerph17197302_

Round 1

Reviewer 1 Report

This study primarily examines whether COVID-19 social confinement exacerbates lower back pain (LBP) in participants from Riyadh (Saudi Arabia). The statistics provided by the authors reveal that a high proportion of professionals suffer from LBP; hence, the importance of this study within a public health context. The manuscript needs more clarity and would also benefit from a more rigorous proofreading. The statistical analyses and the report of results could be clearer. It is evident from the outcomes reported that the COVID-19 lockdown has negatively impacted individuals residing in Riyadh, most importantly exacerbating LBP. Implications for public health of the differences observed are discussed.

Specific comments:

The title could have been better formulated. For instance, increase in pain intensity and prevalence are themselves risks associated with COVID-19 lockdown as the study reveals. Hence, the other associated risk factors could have been included in the title to better reflect what was studied.

It would have been helpful if the authors could insert in the manuscript the questionnaire they employed to better understand the various factors they assessed.

Lines 30-33: Avoid repetition of results in the abstract.

Line 37: I would suggest replacing “health and political authorities” with “public health authorities” and provide a relevant reference.

Lines 40 and 43: These claims relate to specific regions as evidenced from references [2] and [3] and should therefore be stated as such. Otherwise, it would be more appropriate to cite references that support a more global concern.

Line 52: Typo “there”

Line 55: Grammatical mistake “in the dentist”.

Line 56: Do the authors mean male secondary school teachers (reference [12])?

Line 69: “feasible” is not the right adjective.

Line 78: The specified age range for recruitment is rather wide, yet the average age is 35.63 years. Were the participants mostly young? I would like to see more information about the age range and average age of females and males; age groups and the number of participants in each (since age groups are compared).

Lines 171; 174: How was pain intensity assessed? Was it a composite score on the scale 1 to 5? What was the threshold separating low from high pain intensity?

Lines 175-180 and 181-185: Were the three age groups compared together using the repeated measures? If not, what was the rationale for comparing the groups independently?

Line 181: How many participants constituted the three BMI groups?

Line 181: Typo “significantly”.

Line 183: Typo “than”.

Lines 184, 207, 214, 225, 285: Replace “referred” more appropriately with “reported”. Please also check the use of “referred” throughout the manuscript.

Lines 206-208: Do the authors mean “significantly lower LBP intensity” as reported in the Table 2?

Line 210: Typo “significantly”.

There are many results that are reported, which can be a bit overwhelming. Perhaps the authors could devise a more meaningful table (similar to Table 3) where all the significant differences, exact p values and ES values are reported. Then they can make observations about significant differences in the text.

Lines 247-248: It is not clear how this is relevant since the authors measured change in prevalence over a short duration.

Based on their interpretation and discussion of their results, could the authors also discuss the validity and reliability of their measurement?

Reviewer 2 Report

This is an interesting study intended to evaluate the impact of the Covid-19 quarantine on (low back) pain and associated sociodemographic and behavioural factors in adult citizens residing in Riyadh. There are a limited number of population-based studies examine quarantine effects on reported pain levels. As such, the study is an important addition to research in this area. There is much to be encouraged by the scope of collected data and the findings are interesting, but there are some problems concerning the methodology/analysis and interpretation which need to be addressed.

Major points

  1. Methods p.3 Lines 97-98. It is stated that the ‘Questionnaire responses were structured on a scale of whole numbers from 1 to 5’. Does this refer to items concerning pain intensity and/or perceived stress? What were the scale anchors? Was there a timeframe around participants’ reported rating/experience of pain (e.g., in the last 7 days)? More details are needed.
  2. Methods p.4 Lines 108-109. It is stated that ‘Contact with potential study participants was established through the Riyadh municipality forum groups available in social media’ – what forum groups were involved/sampled and how representative are members of the forum groups with respect to the Riyadh municipality population?
  3. Methods p.2 Lines 80-81. While it appears that individuals with chronic disease were excluded from the survey, did this include people with chronic pain conditions (e.g., fibromyalgia) or psychiatric illnesses? More generally, data about the chronicity of (lower back) pain in participating individuals would be helpful.
  4. Results Table 1. A more helpful approach would be to compare frequencies (percentages) before and during lockdown using conventional statistical tests (e.g., McNemar) – that way it could be established whether differences during the lockdown were significant for the measured variables (the percentage change measure is problematic in so much as it is closely linked with baseline percentage and does not best represent magnitude of change).
  5. Results Table 2, Table 3. Univariate analyses indicate that a number of variables were associated with higher LBP intensity, including age, BMI, perceived stress, non-adherence to ergonomic recommendations, prolonged sitting, and insufficient practice of PA. However, some of these factors are likely to covary with each other. A multivariate analytical approach (e.g., generalized linear modelling, linear regression - using bootstrapping to estimate confidence intervals of coefficients in the case of non-normal dependent measure) would better discern the most relevant factors to increased pain post lockdown. To this end, directly examining before-to-during lockdown change scores on intensity and the relationship with corresponding changes in risk factors would be preferable to separately examining relationships before and during lockdown (or alternatively, employ a mixed modelling approach including data on pain and associated factors both before and during lockdown).
  6. Results Table 2, Table 3. Several associations/comparisons concerning pain intensity are reported. When so many variables are correlated/compared - there is a high risk of Type I errors. As such, there is a need for a correction for multiple associations/comparisons (e.g., Bonferroni, False Discovery Rate).
  7. Discussion p.9 Lines 241-243. One of the difficulties in claiming that the point prevalence of LBP increased after lockdown is that it remains unclear if this increase is merely numerical or in fact significant. This should be formally tested and reported accordingly (see point 4 above).
  8. The study was cross-sectional and as such the direction of association between pain intensity and behavioural constructs cannot be determined with any certainty (e.g., onset of LBP or increased LBP intensity could lead to increased stress and/or reduced physical activity rather than always be a consequence of these) – this is important and at the very least be acknowledged in the Discussion.
  9. A key difficulty with this study is that data concerning pre-quarantine behaviours and pain levels was ascertained after the lockdown had begun - pain intensity recall is often unreliable and linked to pain intensity experienced during the time of recall and other traits such as pain-related anxiety (Babel, 2017; Daoust et al., 2017; Feine et al., 1998). Some discussion of this is warranted.

Minor comments

  1. Throughout the manuscript the word ‘referred’ is used – perhaps ‘reported’ may be more appropriate in some of these instances?
  2. Methods Figure 1. While the sample size appears sufficiently large for analyses and the sample size calculation is helpful, the calculation itself is unclear – what is the expected probability of success/failure that feeds into the equation?
  3. Methods Figure 2. How many items were originally considered for expert rating in the development of the questionnaire?
  4. Results Table 2. The text concerning the comparisons detailed in Table 2 consists of many reported p and ES values. These values would be better placed in the Table itself in my view. Also, providing the n values for each subgroup examined in Table 2 would be helpful.
  5. Results Table 2. To the extent that pain intensity scores did not closely approximate a Gaussian distribution and non-parametric tests were employed, presenting medians and interquartile ranges as summary data is more appropriate than means and SDs.
  6. Results Table 3. Small point but the table would be easier to interpret if before and during the quarantine correlations were in separate columns.

References

BÄ…bel P. The influence of state and trait anxiety on the memory of pain. Pain Medicine. 2017 Dec 1;18(12):2340-9.

Daoust R, Sirois MJ, Lee JS, Perry JJ, Griffith LE, Worster A, Lang E, Paquet J, Chauny JM, Émond M. Painful memories: reliability of pain intensity recall at 3 months in senior patients. Pain Research and Management. 2017.

Feine JS, Lavigne GJ, Dao TT, Morin C, Lund JP. Memories of chronic pain and perceptions of relief. Pain. 1998 Aug 1;77(2):137-41.

Round 2

Reviewer 1 Report

Dear Authors,

Thank you for addressing my comments and for improving the manuscript. Thanks again for the opportunity to read your manuscript.

All the best.

Reviewer 2 Report

There has been some improvement of the manuscript according to changes made in response to the recommendations of the reviewers, with greater clarity provided around the methodology/analysis and interpretation. There are some points (below) - once addressed, the manuscript could be published in the IJEPR.

  1. Methods p.3 Lines 95-97. The authors note in their reply that it is stated that the questionnaire responses were structured on a scale of whole numbers from 1 to 5 and refer to the Appendix for details of the terms used on scale anchors. But it would be helpful to provide examples of anchor terms in the main text (e.g., for pain, anchor terms were ‘No pain’ to ‘Extreme pain’).
  2. Methods p.2 Lines 77-79. The clarification about the exclusion of individuals with chronic pain conditions (e.g., fibromyalgia) or psychiatric illnesses is helpful. But a brief note in the Limitations of the Discussion noting that the findings do not provide any information about how people with existing chronic (back) pain were affected is warranted.
  3. Methods p.5 Line 135. ‘Benferroni’ should be ‘Bonferroni’?
  4. Results Table 1. Small point but where p value is reported as .000 this should be <.001.
  5. Results Table 2. To the extent that pain intensity scores did not closely approximate a Gaussian distribution and non-parametric tests were employed, presenting medians and interquartile ranges (IQRs) as summary data is more appropriate than means and IQRs.
  6. Results Table 2, Table 3. The authors have employed univariate analyses to indicate that a number of variables were associated with higher LBP intensity, including age, BMI, perceived stress, non-adherence to ergonomic recommendations, prolonged sitting, and insufficient practice of PA, and appear reluctant to adopt a multivariate analytical approach to better discern the most relevant factors to increased pain post lockdown. Consequently, at a minimum, acknowledgment in the Discussion that some of these variables are likely to covary with each other is needed.
  7. Results Table 3. The addition of a column specifying associated p values is welcome. But, as before, the table would be easier to interpret if before and during the quarantine correlations were in separate columns rather than in rows as at present.
  8. The authors responded in their reply letter to the point concerning the reliability of pain intensity recall, arguing against the possibility that pain intensity recall may have been unreliable and citing relevant research in support of their position. While I’m not entirely convinced that recall was always accurate (as previously noted, pain intensity recall is often linked to pain intensity experienced during the time of recall and other traits such as pain-related anxiety, Babel, 2017; Daoust et al., 2017; Feine et al., 1998), brief consideration of this matter is warranted in the Discussion.

References

BÄ…bel P. The influence of state and trait anxiety on the memory of pain. Pain Medicine. 2017 Dec 1;18(12):2340-9.

Daoust R, Sirois MJ, Lee JS, Perry JJ, Griffith LE, Worster A, Lang E, Paquet J, Chauny JM, Émond M. Painful memories: reliability of pain intensity recall at 3 months in senior patients. Pain Research and Management. 2017.

Feine JS, Lavigne GJ, Dao TT, Morin C, Lund JP. Memories of chronic pain and perceptions of relief. Pain. 1998 Aug 1;77(2):137-41.
